# Solvent-Free Mechanochemical Preparation of Metal-Organic Framework ZIF-67 Impregnated by Pt Nanoparticles for Water Purification

Mahya Afkhami-Ardekani [1], Mohammad Reza Naimi-Jamal [1],*, Samira Doaee [2] and Sadegh Rostamnia [2],*

[1] Research Laboratory of Green Organic Synthesis and Polymers, Department of Chemistry, Iran University of Science and Technology (IUST), Tehran P.O. Box 16846-13114, Iran

[2] Organic and Nano Group (ONG), Department of Chemistry, Iran University of Science and Technology (IUST), Tehran P.O. Box 16846-13114, Iran

* Correspondence: naimi@iust.ac.ir (M.R.N.-J.); rostamnia@iust.ac.ir (S.R.)

**Abstract:** In this study, the crystalline metal-organic framework (MOF) ZIF-67 was obtained using the solvent-free ball milling method, which is a fast, simple, and economical green method without the need to use solvents. Using the impregnation method, platinum metal ions were loaded in the MOF cavities. Various descriptive methods have been used to explain the prepared Pt@ZIF-67 compound, such as FTIR, BET, TEM, SEM, EDS, XRD, TGA, and ICP. Based on this, the results showed that Pt nanoparticles (0.26 atom%) were located inside the pores of ZIF-67. In addition, no evidence supports their accumulation on the MOF surface. The efficiency of Pt@ZIF-67 was approved in the reduction of toxic and harmful nitrophenol compounds in water. The results showed that the removal of 4-nitrophenol in aqueous medium was successfully achieved with 94.5% conversion in an optimal time of 5 min with the use of $NaBH_4$, and catalyzed by Pt@ZIF-67. Additionally, the Pt@ZIF-67 was recoverable and successfully tested for five qtr runs, with reasonable efficiency.

**Keywords:** ball-milling; platinum nanoparticles; metal-organic framework (MOF); nitrophenol reduction; ZIF-67; impregnation method





## 1. Introduction

Ball mill (BM) is one of the mechanical methods widely used for pulverization and conversion of solids into fine particles [1,2]. When raw materials get stuck between the bullets and are crushed by rotation, because of the potential energy release, some surface changes would occur, including breakage of bonds. Therefore, in this process, not only is the particle size reduced, but it also leads to reactions through the formation of active sites [3]. To achieve high quality and pure products, it is essential to control different parameters, such as the type of substrates, the diameter and the density of the milling balls, the rotation speed, and the milling time [4]. Since this method is environment-friendly and cost-effective, it is widely applied in synthesizing organic, mineral, and metallurgical nanomaterials [5]. Due to the advantages such as short reaction time, the absence of solvent, and lack of a requirement to external heating, this method has made significant progress in synthesizing metal-organic frameworks. Today, ball mill has attracted much attention to nanomaterials, leading to considerable changes in this area [6–8]. Metal-organic frameworks (MOFs) are the polymers of metal ions and carbon-based ligands, formed to create three-dimensional crystal structures [9]. Because of the exclusive periodic pore building and brilliant properties, these structures have been used in a variety of fields such as adsorption [10,11], separation [12,13], catalysis [14–16], sensing [17], electromagnetism [18], and medicine [19,20]. Due to their pore structure tunability, the high porosity and high surface area, scientists have paid special attention to MOFs as adsorbents in aqueous solutions [21]. Due to low contamination, low reaction time, and high capability

for large-scale synthesis, ball mill is particularly suitable for producing MOFs. In recent years, many important advances have been made in MOF-based derivatives [22–25]. It is noteworthy that by using different nanoparticles, such as metal nanoparticles (MNP), in the matrix of metal-organic frameworks, the potential applications of these derivatives can be increased [26–29]. For example, imidazole zeolite frameworks (ZIFs) are a new type of metal zeolite with complex transfer of metal ions using an imidazole ligand. This compound has some specific properties, such as great strength and porosity, mainly used as hard mold in the preparation of carbon materials [30]. Due to its high thermal, chemical, and porosity stability, ZIF-67 is growing in prominence [31]. Moreover, water filtration was a common application for ZIF-67. The exponential development of industries and the increase in the industrial and agricultural wastewater have brought various harmful and dangerous chemical pollutants into the environment [32,33], such as pesticides [34], pathogenic microbes [35], and explosives [36]. Amongst these, 4-nitrophenol has received much consideration due to its rapid negative effect on the human nervous system as well as showing high polarity and solvability [37,38]. Different methods have been developed to control this organic pollutant, such as adsorption [39,40], photocatalytic destruction [41,42], and catalytic destruction (including reducing and oxidative degradation) [43–45], among which catalytic reduction is a method for reducing 4-nitrophenol (4-NP) to 4-aminophenol (4-AP), using sodium borohydride as the reducing agent [46,47]. Noble metal nanoparticles (NPs), such as gold, platinum, and palladium, are suitable options, due to their superior consistency and the great surface-to-volume ratio [48–52]. In this paper, the method of producing the ZIF-67 organic metal framework and its nano-scale impregnation with platinum metal as well as applying it in eliminating the toxic 4-NP from water have been studied and reported.

## 2. Results and Discussion

The ZIF-67 metal-organic framework was synthesized through ball milling a mixture of cobalt nitrate hexahydrate and 2-methylimidazole without the addition of solvent (Figure 1). The platinum metal was mixed and the obtained Pt@ZIF-67 was used as a catalyst in the reduction of 4-NP in the presence of the $NaBH_4$.

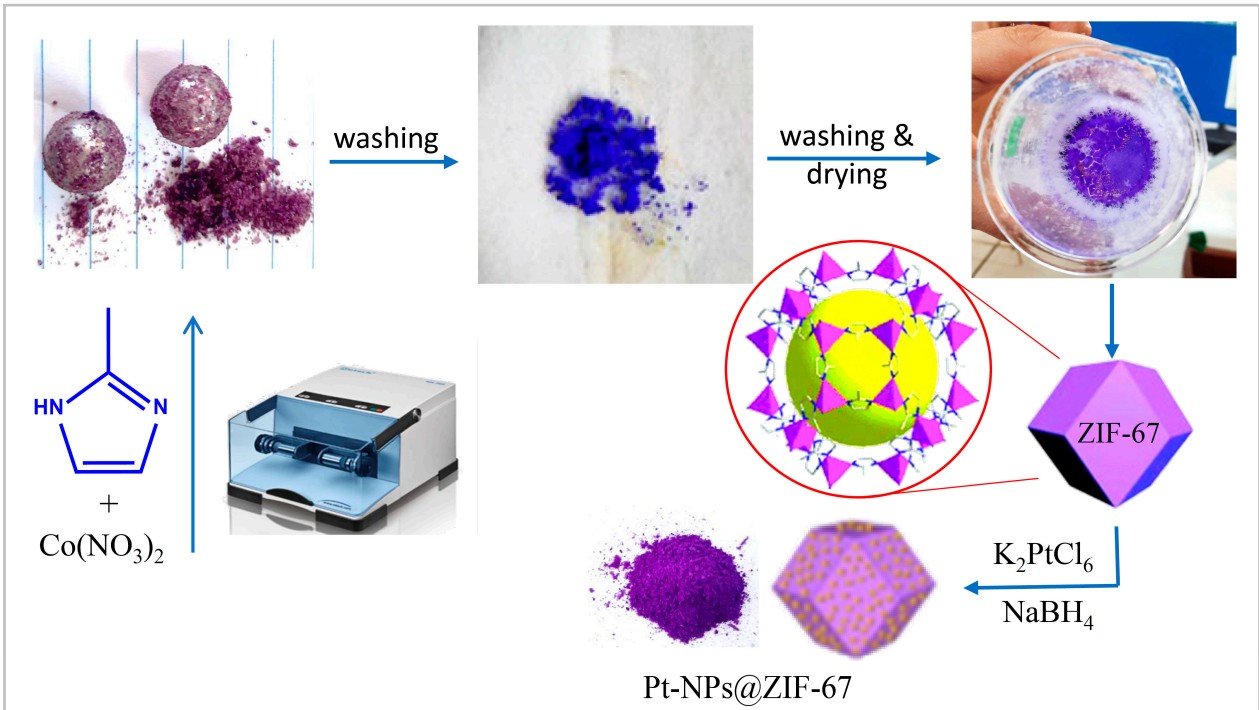

**Figure 1.** Schematic BM mediated synthesis of Pt@ZIF-67.

The FT-IR spectra of the synthesized ZIF-67 and Pt@ZIF-67 are reported in Figure 2. In the FT-IR spectrum of the ZIF-67 structure, the peak of 2922.8 cm$^{-1}$ indicates the stretching vibration of aliphatic C-H of methyl imidazole. The peaks at 1633 and 1416 cm$^{-1}$ are related to the C=N and C=C groups of imidazole, respectively. In addition, the peak of 3439 cm$^{-1}$ is related to the hydroxyl groups (O-H) and the peak of 425 cm$^{-1}$ corresponds to the vibration of the Co-N bond [53]. Due to the low concentration of Pt nanoparticles in the ZIF-67 structure's cavities, the synthetic Pt@ZIF-67 structure does not exhibit a distinct peak for Pt nanoparticles in the FTIR.

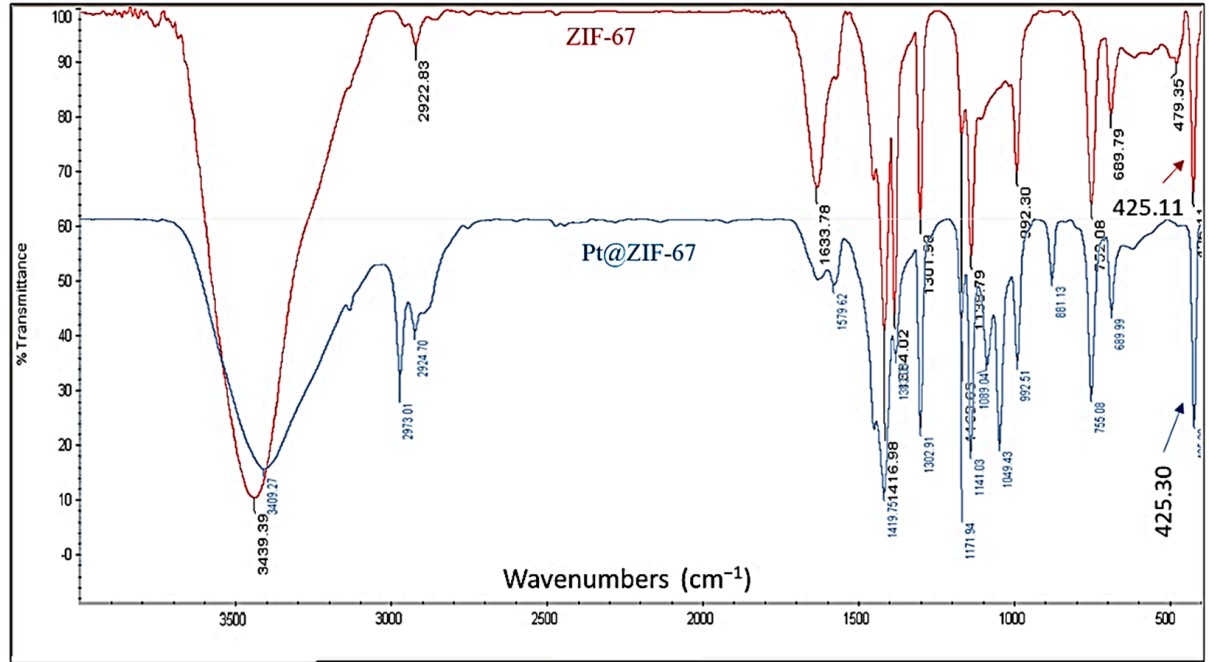

**Figure 2.** Comparison of FTIR spectra of ZIF-67 and Pt@ZIF-67.

To investigate the crystal structure of the synthesized Pt@ZIF-67, X-ray powder diffraction (XRD) was used. The diffraction peaks in the XRD patterns of the synthesized structures of ZIF-67 and Pt@ZIF-67 are shown in Figure 3. Comparing the resulting patterns, it was found that all the peaks related to the ZIF-67 structure are also present in the Pt@ZIF-67 structure, indicating that the platinum metal in the ZIF-67 structure did not damage the structure of this metal-organic framework. The patterns of the ZIF-67 had many obvious diffraction peaks at 2θ = 7.42°, 10.46°, 12.81°, 14.79°, 16.53°, and 18.12°. Examining the XRD pattern of the simulated ZIF-67 model (Figure 3a), it was found that the ZIF-67 and Pt@ZIF-67 did not show any significant decrease in crystallinity. In addition, in the XRD model, the synthesized structure of Pt@ZIF-67 does not show a clear peak for Pt nanoparticles, resulting from the low concentration of Pt nanoparticles in the cavities of ZIF-67 structure.

The surface area and the whole pore volume of ZIF-67 and Pt@ZIF-67 were studied using N$_2$ physical adsorption experiments (Figure 3b and Table 1). The N$_2$ adsorption-desorption isotherms of ZIF-67 and Pt@ZIF-67 are type I, indicating their microporous properties. The BET surface area of ZIF-67 and Pt@ZIF-67 were 1138 and 648.89 m$^2$ g$^{-1}$, respectively. Moreover, the total pore volume was 0.5235 and 0.302 cm$^3$ g$^{-1}$, respectively. Obviously, for Pt@ZIF-67, the surface area and total pore volume are less than those for the parent ZIF-67, approving the successful immobilization of platinum nanoparticles in the ZIF-67 pores. Pore size distribution experiments show that ZIF-67 and Pt@ZIF-67 are micro pore types, with average diameters of 1.84 and 1.86 nm, respectively. According to the stated BET analysis results, there is no discernible difference between the average diameter of ZIF-67 and Pt@ZIF-67 structures, or, to put it another way, nothing out of the ordinary has happened.

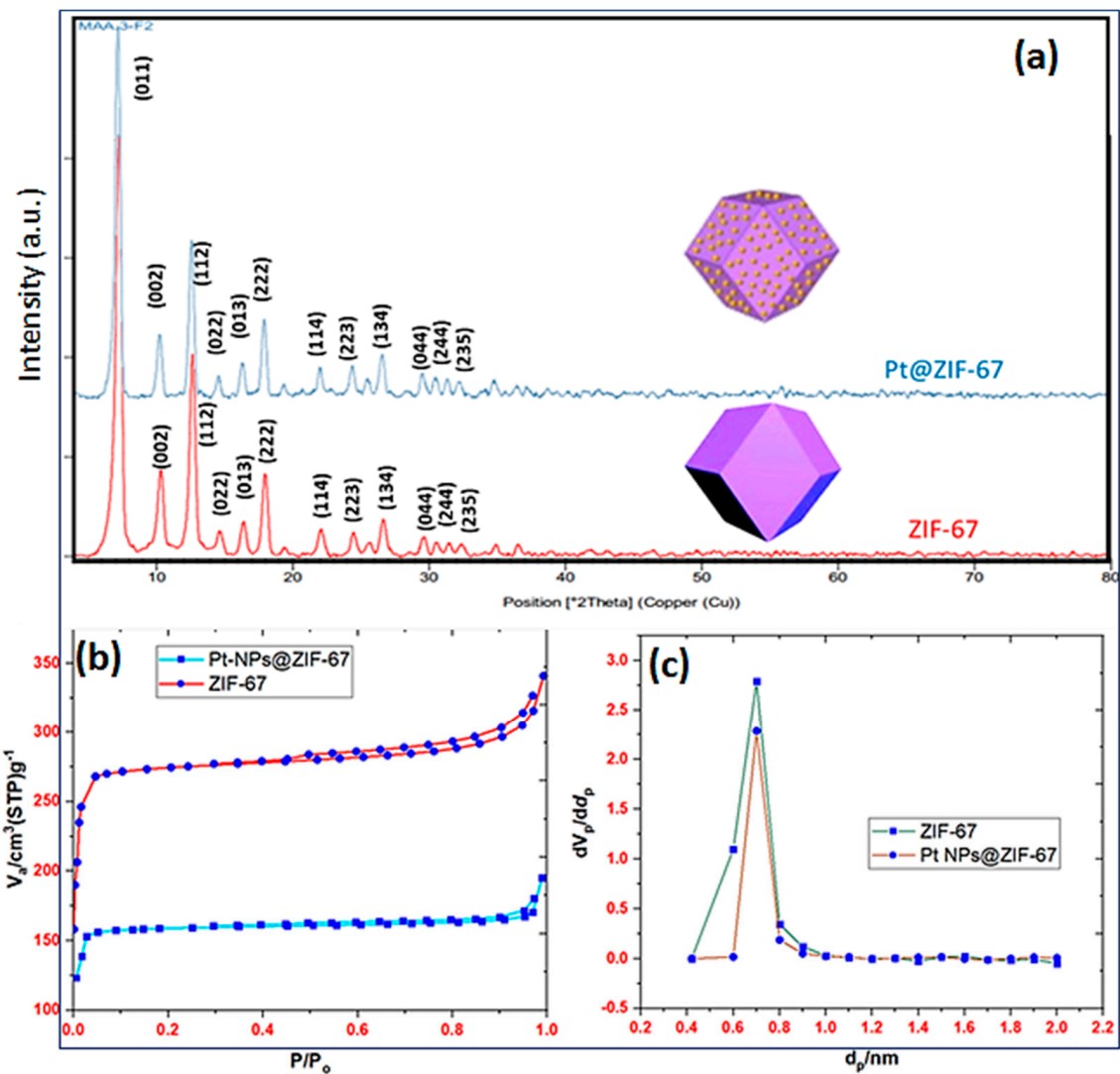

**Figure 3.** (**a**) ZIF-67 and Pt@ZIF-67: XRD pattern of (red) ZIF-67; (blue) Pt@ZIF-67. (**b**) The $N_2$-adsorption-desorption isotherm of and (**c**) the pore size distribution of the ZIF-67 and the Pt@ZIF-67.

**Table 1.** Specific surface area (SBET), diameter pore, and total pore volume.

| Sample | BET Surface Area $(m^2\ g^{-1})$ | Average Diameter (nm) | Total Pore Volume $(cm^3\ g^{-1})$ |
|---|---|---|---|
| ZIF-67 | 1138 | 1.84 | 0.5235 |
| Pt@ZIF-67 | 648.89 | 1.86 | 0.302 |

To observe the morphology of ball mill produced ZIF-67, FESEM and TEM were performed. The FESEM images show spherical morphologies for both primary ZIF-67 and Pt@ZIF-67 particles, with a nano-size distribution between 50 and 100 nm. The TEM image of Pt@ZIF-67 showed good distribution of Pt nanoparticles in the MOF structure. The dark dots in the TEM pictures provided evidence that platinum was present in the Pt@ZIF-67 (Figure 4).

The EDS spectroscopy results and SEM element mappings support well the presence of the elements N, C, Co, and Pt (Figure 5). Additionally, Table 2 shows that 3.18 wt.% (0.26 A.%) of platinum nanoparticles is contained in the Pt@ZIF-67. Furthermore, we discovered that the particles are equally dispersed across the surface by examining the mapping pictures of the elements extracted from the SEM image of Pt@ZIF-67.

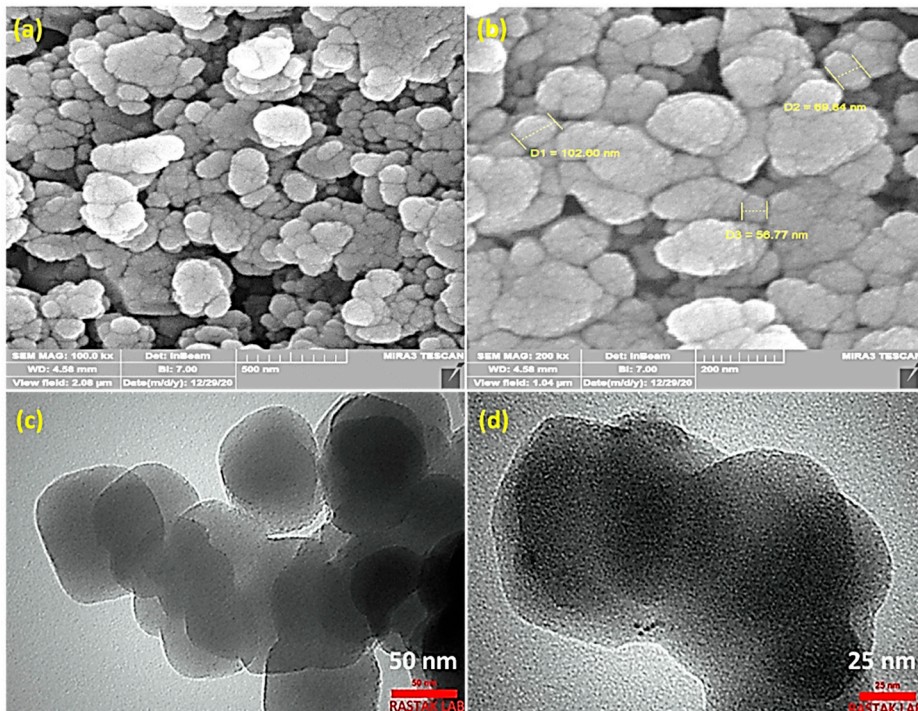

**Figure 4.** (**a**) The SEM, (**c**) TEM of ZIF-67, (**b**) The SEM, and (**d**) TEM of Pt@ZIF-67.

The thermal stability of the synthesized ZIF-67 and Pt@ZIF-67 was evaluated by thermal gravimetric analysis (TGA) (Figure 6). The thermal behavior of both MOFs is essentially the same, approving the similarity between chemical composition and crystal structure. Both are stable at temperatures up to 300 °C, and lose 60% of their weight at higher temperatures up to 700 °C. This confirms the presence of similar organic moieties in their structure.

The catalytic reduction of the toxic compound 4-NP into 4-AP was performed by adding the synthesized catalyst and in the presence of $NaBH_4$ as the reducing agent. This way, metal particles in the supporting materials can accelerate the transfer of electrons from $NaBH_4$ to the 4-NP electron receptor and ultimately reduce the hydrogenation of 4-NP to 4-AP. Considering enough runs for the given model can help in evaluating and identifying the most important variables. Response Surface Methodology (RSM) is a combination of mathematical and statistical procedures to determine the optimal range of parameters. Even in complex interactions, this range can be used to determine and evaluate the relative importance of the parameters. Analysis of the results was performed based on the ANOVA. In this modeling method, by fitting the first or second polynomial equation, the experimental answer is obtained and used in the relevant experimental design. The analysis of variance (ANOVA) is then performed. As a result, the approved model of the surface response modeling method would be presented as a three-dimensional diagram as well as a contour diagram. According to the response function, the best operational and experimental conditions for the process are determined. In general, RSM analysis uses a second-order polynomial model for the expected response ($Y$):

$$Y = \beta_0 + \sum_{i=1}^{k} \beta_i X_i + \sum_{i=1}^{k} \beta_{ii} X_i^2 + \sum_{i=1}^{k} \sum_{j=1}^{k} \beta_{ij} X_i X_j + \varepsilon$$

where $\beta_0$ is the expression of the point of intersection, $\beta_i$ the constant of the linear factor, $X_i$ and $X_j$ are the parameters assigned to the factors *i* and *j*, $\beta_{ii}$ and $\beta_{ij}$ are the coefficients of the square factor and the interaction factor, and *k* is the number of rate factors [54,55].

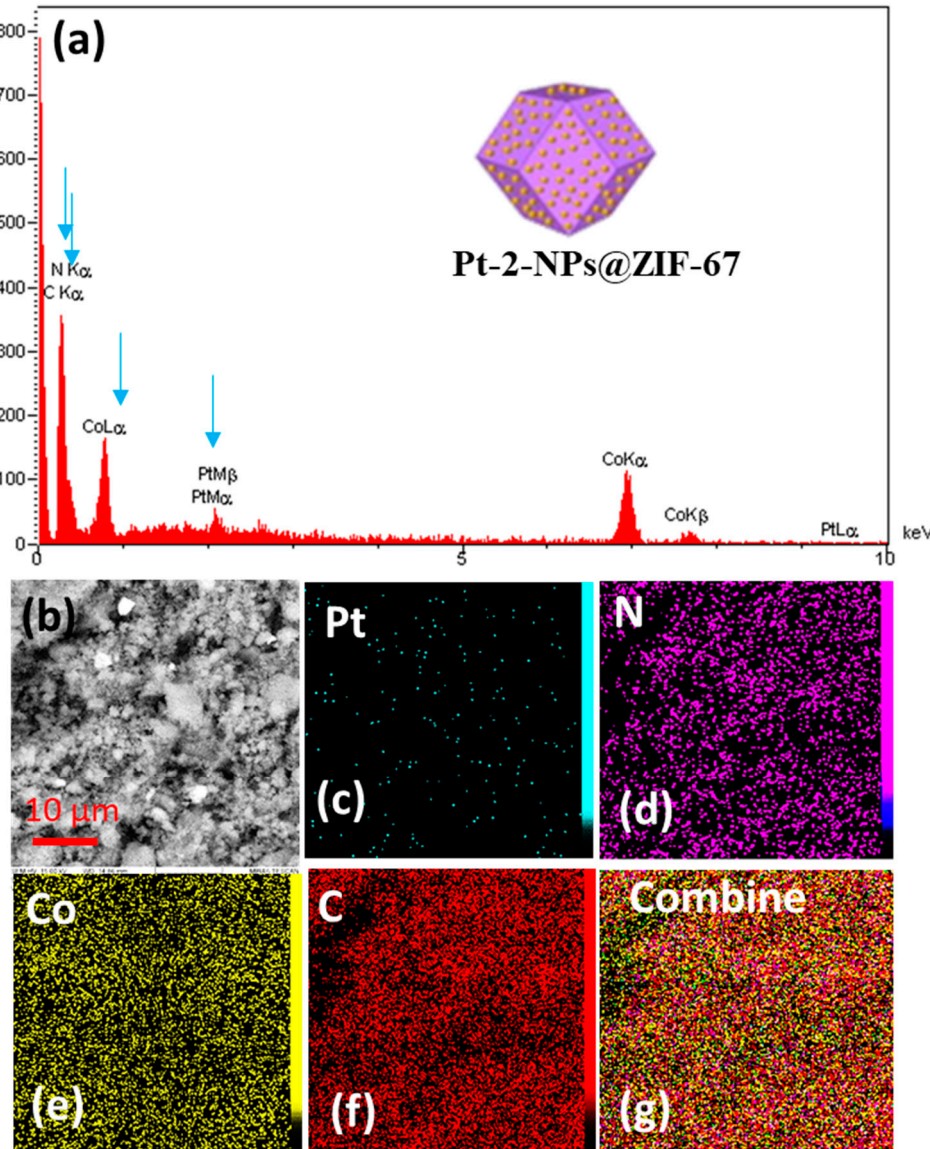

**Figure 5.** Pt@ZIF-67: (**a**) The SEM-EDS spectra and (**b–g**) SEM-mapping.

**Table 2.** Weight percentage of elements from EDS analysis.

| Element | Line | Int | Error | K | Kr | W% | A% | ZAF | Ox% | Pk/Bg |
|---------|------|------|----------|--------|--------|--------|--------|--------|------|-------|
| C | Ka | 345.6 | 187.3370 | 0.4099 | 0.1759 | 38.51 | 51.78 | 0.4569 | 0.00 | 67.97 |
| N | Ka | 91.6 | 187.3370 | 0.1468 | 0.0630 | 36.32 | 41.87 | 0.1735 | 0.00 | 43.69 |
| Cl | Ka | 20.6 | 1.8728 | 0.0063 | 0.0027 | 0.32 | 0.15 | 0.8383 | 0.00 | 2.63 |
| Co | Ka | 313.0 | 0.6091 | 0.3972 | 0.1705 | 21.67 | 5.94 | 0.7868 | 0.00 | 25.37 |
| Pt | La | 2.2 | 0.6091 | 0.0398 | 0.0171 | 3.18 | 0.26 | 0.5364 | 0.00 | 2.30 |
| | | | | 1.0000 | 0.4292 | 100.00 | 100.00 | | 0.00 | |

The experimental layout was determined as a function of the most essential factors, including reducing agent concentration (A), catalyst value (B) and 4-NP concentration, for reaching the optimal conditions of eliminating 4-NP. The destruction percentage of each pollutant was considered the parameter response (Y) and each reaction's total reduction time was calculated as the time parameter. Finally, 20 experiments were carried out in the same conditions (Table 3). For more information, data were collected as separated files (see Supporting Information (Tables S1–S3)).

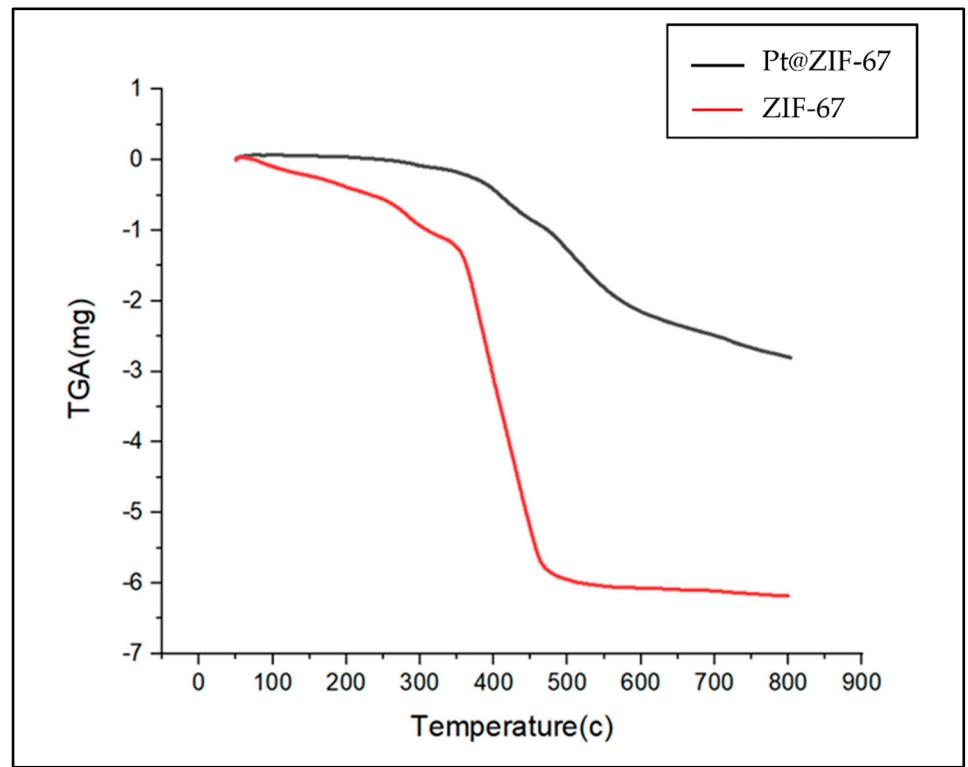

**Figure 6.** The TGA curve of ZIF-67 (bottom), Pt@ZIF-67 (top).

**Table 3.** Results from experiments designed by RSM.

| Std. | Run | Response (R) | Time (min) |
|---|---|---|---|
| 1 | 9 | 26.48 | 10 |
| 2 | 11 | 69.48 | 8 |
| 3 | 13 | 70.56 | 8 |
| 4 | 2 | 89.78 | 6 |
| 5 | 17 | 39.27 | 7 |
| 6 | 20 | 50.58 | 14 |
| 7 | 10 | 85.56 | 5 |
| 8 | 15 | 79.30 | 7 |
| 9 | 4 | 53.54 | 4 |
| 10 | 18 | 93.07 | 6 |
| 11 | 1 | 8.37 | 10 |
| 12 | 12 | 98.58 | 4 |
| 13 | 6 | 90.80 | 11 |
| 14 | 8 | 62.81 | 17 |
| 15 | 14 | 82.11 | 5 |
| 16 | 16 | 96.92 | 9 |
| 17 | 19 | 79.71 | 6 |
| 18 | 5 | 82.50 | 5 |
| 19 | 7 | 83.97 | 11 |
| 20 | 3 | 92.87 | 6 |

In general, based on Figure 7a–c, it was found that increasing the concentration of $NaBH_4$ leads in maximum response, but at lower levels, minimum response would be observed. The maximum response range for $NaBH_4$ is from 0.0015 to 0.002 M. Considering the maximum concentration of $NaBH_4$, the influence of other parameters can be achieved. Consumption of the Pt@ZIF-67 catalyst should be increased in the range of 0.006 to 0.01 g and be decreased in the lower amounts. In addition, in the concentration range of 0.001 to 0.0015 M of 4-nitrophenol, the model response is maximal and at higher levels the response

is minimal. Therefore, increasing the concentration of NaBH$_4$ in the solution contributes in maximum response. On the other hand, promoting the 4-NP concentration leads in minimum response in the model.

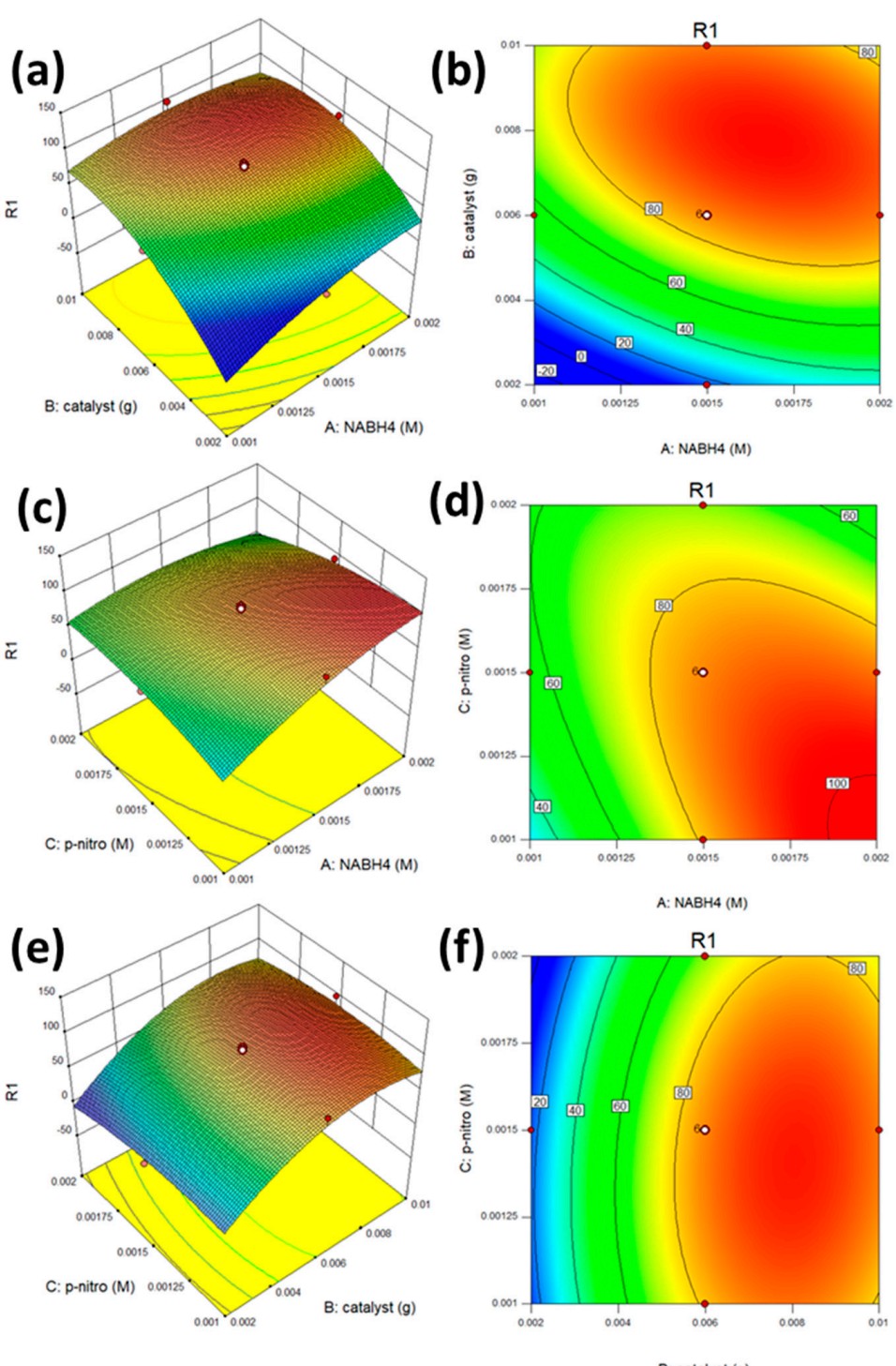

**Figure 7.** (**a**–**c**), and (**b**–**f**); 3D and 2D response areas, respectively, as a result of the simultaneous effects of binary parameters on the response size.

The results showed that the reduction of 4-nitrophenol increased with time. In addition, a test predicted by the final model was designed to be optimized according to the proposed test conditions. Conforming the predicted data and one of the 20 samples tested (Table 4 and Figure S1), it was possible to match 97.3% of the actual value tested and the total

predicted value of the model, which was calculated in the Design Expert10 software to optimize the model (see Supporting Information).

**Table 4.** The results obtained and their conformity of the expected values of the model.

| Number | $NaBH_4$ (M) | Catalyst (g) | 4-NP (M) | R (Accuracy) | Desirability |
|---|---|---|---|---|---|
| 1 | 0.0015 | 0.006 | 0.0015 | 86.44 | 1.000 |

The concentration of cobalt released in the solution after the completeness of the reaction was equal to 0.28 ppm, which is equivalent to roughly 0.5% leaching, and the concentration of platinum released in the solution after the reaction was also equal to 0.054 ppm, which is equivalent to less than 1% leaching, according to an ICP Optical Emission (ICP-OES) analysis of the nitrophenol reduction reaction solution under optimal conditions. Additionally, the recycling test for Pt@ZIF-67 was carried out for five consecutive runs of the model reaction. The results showed that the catalyst can be recovered and reused several times, although with some minor decrease in its activity after each run. (Figure 8).

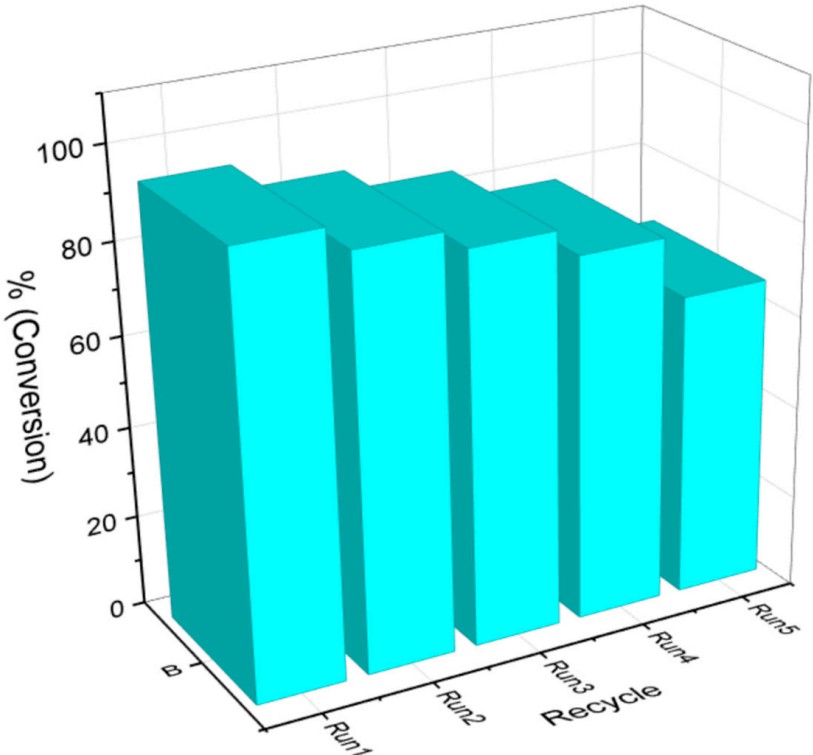

**Figure 8.** Reusability study of the Pt@ZIF-67 catalyst.

In addition, a brief comparison of our investigation was provided with earlier studies on 4-nitrophenol reduction utilizing different catalysts and $NaBH_4$. According to this, we listed various kinds of catalysts in terms of time, efficiency %, and reaction conditions, as summarized in Table 5. In particular, it can be expressed that the catalytic activity obtained by our catalyst is prominent when compared with the previously mentioned catalysts.

Reviewing the process by which 4-NP reduces on the surface of metals using $NaBH_4$ (as a reductant), 4-NP first deprotonates to 4-aminophenolate, then moves toward the metal's surface (in this case, Pt), accepts the electron, and then transforms to the reduced state. The donated electron from the catalyst is provided by $NaBH_4$. A possible scheme for the conversion of 4-NP to 4-AP is displayed in Figure 9.

**Table 5.** Summary of materials that can be used for the reduction of 4-NP using $NaBH_4$ previously reported.

| Catalyst | Reduction Condition | Efficiency (%) | Time (min) | Refs. |
|---|---|---|---|---|
| Pd-PBL | Cat. (1 mol %)<br>4-NP ($10^{-3}$ mmol)<br>$NaBH_4$ ($10^{-4}$ mmol) | 98.3 | 32 | [56] |
| Bentonite clay supported Fe NPs | Cat. (10 mg/L)<br>4-NP (0.2 mM)<br>$NaBH_4$ (0.2 M) | 96.8 | 20 | [57] |
| NPC | Cat (30 mg)<br>4-NP (30 mL, 0.015 mmol)<br>$NaBH_4$ (1.5 mmol) | 100 | 20 | [58] |
| Pt@ZIF-67 | Cat (6 mg)<br>4-NP (1.5 mM)<br>$NaBH_4$ (1.5 mM) | 96.5 | 5 | This work |

PBL: Phenylene-Bridged Bis(thione) Ligands; NPs: Nanoparticles; NPC: Nitrogen and Phosphorus co-doped Carbon-based metal-free.

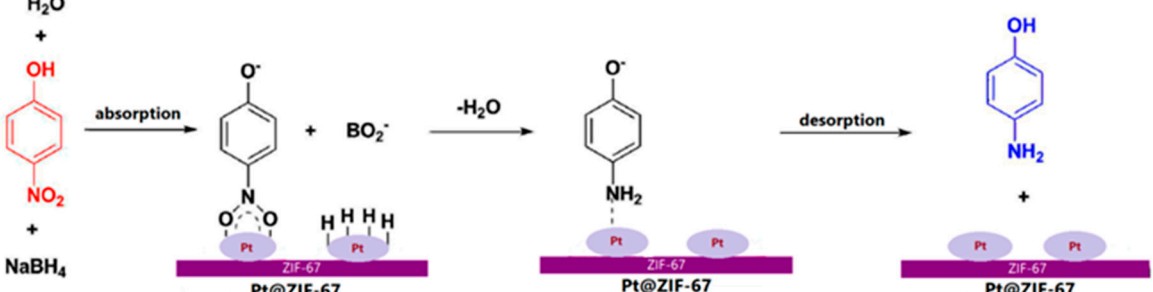

**Figure 9.** The mechanism of reduction reaction 4-nitrophenol by Pt@ZIF-67.

## 3. Materials and Methods

Cobalt (II) nitrate hexahydrate, 2-methylimidazole ($C_4H_6N_2$), potassium hexachloroplatinate ($K_2PtCl_6$), sodium borohydride ($NaBH_4$), methanol, and ethanol were purchased from Merck and Fluka and used directly without any purification. Brunauer–Emmett–Teller (BET) specific surface area, volumetric nitrogen adsorption/desorption curves, and pore volume were recorded on a Belsorp mini and Finetec appliance at 77 K. The surface morphology of the samples was evaluated with Field Emission Scanning Electron Microscopy (FESEM, TE-SCAN Device model MIRA III made in the Czech Republic). The X-ray diffraction energy measurement and mapping of the samples were measured by Ultima iv-Rigaku X-ray diffractometer (made in Japan) with Cu K$\alpha$ radiation ($\lambda$ = 1.5406 Å) between 0 and 80° (2θ). Fourier-transform infrared spectroscopy (FT-IR) spectra were recorded on an AVATAR-Thermo FTIR (made in the U.S.) range of 400−4000 cm$^{-1}$. For thermogravimetric (TGA) analysis under $N_2$, STA 504-1700-50 °C thermal analyzer was used. UV–Vis spectra were applied with a wavelength ranging from 200 to 550 nm using a Perkin–Elmer UV–Vis spectrophotometer. The ball mill was a Retch MM400 with stainless steel jar and balls.

### 3.1. Ball Mill Mediated Preparation of ZIF-67

In a characteristic synthesis, 0.225 g (0.77 mmol) Co(NO$_3$)$_2$·6H$_2$O and 0.622 g (0.76 mmol) 2-methylimidazole ($C_4H_6N_2$) were mixed, then placed in a ball mill and ground at a frequency of 18 Hz for 10 min (optimized time and frequency of BM). Afterward, the precipitates were washed several times with methanol, then collected by centrifugation. They were put in an oven at 60 °C for 12 h to get solid ZIF-67. When working with a ball mill, it is very important to pay attention to the two points of time and frequency of rotation. Due to the number of repetitions of the synthesis, we found that in the synthesis of this MOF, if the time and frequency of rotation are less than the specified value, the

raw materials are not mixed well with each other, and we will not get pure products. At a higher time and frequency, due to the long vibration and long duration of the milling, the wear of the ball will lead to the synthesis of products with some swarf and, as a result, low efficiency.

### 3.2. Preparation of Pt@ZIF-67

To load the platinum metal into the nanopores of ZIF-67, we used the impregnation method. In this method, first, the 0.1 g of MOF prepared from the previous step was poured on a watch glass and spread through by a glass rod. Then 0.006 g of $K_2PtCl_6$ in 2 mL of DI-water was added drop by drop. while stirring with a glass rod until it dries. Then the powder was washed with ethanol to get solid $Pt^{+4}$@ZIF-67. In addition, 0.006 g of $NaBH_4$ in 2 mL of DI-water was used as reducing agent for generation of Pt nanoparticles from $Pt^{+4}$ which loaded on ZIF-67 MOF. The impregnated method was also used for this step, added as described in the previous step. In this way, the platinum ions were loaded on the desired MOF and reduced by $NaBH_4$ in situ. Finally, ethanol was added and the mixture was centrifuged, and the solid was washed with ethanol and dried in an oven. Based on the EDS analysis, the amount of Pt loaded in the ZIF-67 cavities was 3.18 wt.%.

### 3.3. Catalytic Activity of $NaBH_4$

Pt metal nanoparticles in supported ZIF-67 materials can accelerate electron transfer from borohydride to the 4-nitrophenol electron acceptor and reduce 4-nitrophenol to 4-aminophenol. Catalytic reduction of 4-nitrophenol to 4-aminophenol in an aqueous solution was performed in the presence of $NaBH_4$ reducing agent at room temperature. At this stage, according to the values and concentrations provided by the surface response modeling (RSM) method and model analysis by ANOVA analysis, the nitrophenol reduction process was performed (see Supporting Information). Finally, 20 experiments were designed and at each stage, after adding each substance including 4-NP, $NaBH_4$, and Pt@ZIF-67 catalyst in predetermined concentrations with a constant volume of 25 mL of distilled water serving as the reaction solvent, until the reduction reaction ended and the solution's color changed, sampling was carried out every minute. Finally, at one-minute intervals, sampling from the solution was done and analyzed by Ultraviolet-Visible spectroscopy (UV-Vis). After observing complete reduction, the color of the solution was changed from yellow to colorless. In addition, to evaluate the response value of the designed model based on the tested values, the results of UV-Vis spectroscopy analysis of each sample were analyzed and analyzed in surface response modeling through Design Expert 10 software.

### 4. Conclusions

In this study, the Pt@ZIF-67 was prepared by ball milling, as a fast, scalable, and economical environmentally friendly method, with no solvent at room temperature. Anodized platinum was immersed into the MOF structure of the pores, and then the final compound was used in the reduction of 4-nitrophenol in the presence of $NaBH_4$. Although the amount of Pt in the MOF structure was small (3.18 wt.%), it can be considered an important step in the reduction of 4-nitrophenol. In addition, the resulting MOF showed good crystallinity, so it can maintain this property after the saturation process. The results show that Pt@ZIF-67 has good catalytic performance due to the reduction of nitrophenol mixes, leading to the removal of these toxins.

**Supplementary Materials:** The following supporting information can be downloaded at: https://www.mdpi.com/article/10.3390/catal13010009/s1, Figure S1: The results obtained and conformity the expected values of the model.; Table S1: Experiments designed 20 performances with 3 variables. Table S2: Table S2. Standard Deviation Analysis (ANOVA) of data obtained for nitro reduction rate. And Table S3: Table S3. Data and the values predicted by the model.

**Author Contributions:** M.A.-A., Conceptualization, investigation, methodology, writing—original draft; M.R.N.-J., supervision, reviewing, and editing the original draft; S.D., experimental design, theoretical calculations; S.R., consultation, reviewing, and editing the original draft. All authors have read and agreed to the published version of the manuscript.

**Funding:** This research was partially funded by Iran University of Science and Technology.

**Data Availability Statement:** Not applicable.

**Conflicts of Interest:** The authors have no relevant financial or non-financial interest to disclose.

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
