# Peer review of "Solvent-Free Mechanochemical Preparation of Metal-Organic Framework ZIF-67 Impregnated by Pt Nanoparticles for Water Purification"

_catalysts, doi:10.3390/catal13010009_

Round 1

Reviewer 1 Report

1.      The preparation of MOFs via Ball milling process is quite normal. With Pt and NaBH4 as catalyst and reduction agents, the reduction of 4-NP is also normal in the wastewater treatment. What is the novelty of this manuscript to combine these two procedures?

2.      In the FTIR section, how would the spectra change after the loading of Pt on ZIF-67?

3.      Please enhance the discussion especially the changes and different between ZIF-67 and Pt@ZIF-67. The authors should avoid providing the results without any analysis.

4.      Why the average pore diameter become larger after the loading of Pt? If Pt species were immobilized in the pores, the pore size should become small. This result is quite strange.

5.      The dispersion of Pt species, which would tested via chemical sorption of hydrogen, is another efficient parameter for materials. The authors could provide this result to certify their conclusion on the immobilization in pores and no XRD peaks for Pt species.

6.      How about the leaching of cobalt and platinum from Pt@ZIF-67 in aqueous solution? Authors should provide relevant data under optimal conditions.

7.      Authors should also provide the regeneration experiments and results of Pt@ZIF-67 for reduction of 4-NP.

8.      Where can the Supplementary materials be found? The uploaded non-published material is just the cover letter.

Author Response

Please find attached our responses.

Reviewer 2 Report

Manuscript ID: catalysts-1994742

The manuscript entitled, “Solvent-free mechanochemical preparation of metal-organic framework ZIF-67 impregnated by Pt nanoparticles for water purification” is a good and interesting work. Authors developed a cost effective method to synthesize ZIF-67 MOF materials as environmental remedy to degrade Nitrophenol. Although work is transparent and can be published but I had selected major revision to enhance the novelty and better readership. I believe it will not take a long for the authors to work on this revision. My comments are,

1.      Abstract is too short, no any data about characterization and application was provided in abstract. Qualitative and quantitative should be added in abstract to increase the interest of researchers.

2.      In introduction part add recent data on MOFs to increase the novality and readership of this article. Should cite these recent articles of MOFs, Surfaces and Interfaces 34 (2022) 102324, Microchemical Journal 164 (2021) 105973.

3.      No data about ZIF-67 and its properties that why authors choose it for synthesis and utilization as nitrophenol removal. Should be provided. Suggested to read these articles on ZIF-67 and cite, Applied Clay Science 190 (2020) 105564, Surfaces and Interfaces 25 (2021) 101261.

4.      In introduction section line 60-63. Briefly describe what you are going to synthesize and what will be the expected results. For more clarification read this article Materials Science and Engineering B 273 (2021) 115417.

5.      Fig. 2 is not clear, should provide clearer Fig.

6.      Author should discuss what happened with FTIR peaks after adding Pt.

7.      Results and discussion should be one heading not separate

8.      Characterizations should be more discussed (SEM, TEM)

9.      Suggested to draw TGA in single Fig using Origin software

10.  Results of removal of NP should be compared with reported literature.

11.  Proposed treatment mechanism should be provided.

12.  Reusability test should be carried out

13.  In conclusion section results of characterizations and removal of NP should be added

Author Response

Please find attached our responses.

Round 2

Reviewer 1 Report

After reading the revised version, I still felt the manuscript lacks clearly novelty. In addition, the manuscript did not do enough data analysis and discussion. The response was also insufficient. Therefore, I cannot agree to accept manuscript in this journal.

Author Response

Response to reviewers

Reviewer #1:

Manuscript ID: catalysts-1994742

Q:  After reading the revised version, I still felt the manuscript lacks clearly novelty. In addition, the manuscript did not do enough data analysis and discussion. The response was also insufficient. Therefore, I cannot agree to accept manuscript in this journal.

Response: Thanks a lot for your time and reading of our manuscript. About the novelty of the work, we think both the solvent-free ball-milling ZIF-67 synthetic part and nitrophenol reduction part have novelty and they have enough borderline sciences. For the first time, the ball mill (BM) synthetic sustainable method was used for the solid-state synthesis of ZIF-67 at room temperature. In our method, we optimized the milling frequency and the time of milling. Additionally, this research differs from prior studies on the reduction of nitro phenol in the way that a very small amount of platinum metal was utilized to create the catalyst (we compared our data with literature survey-Please see Table 5).

As can be seen in main reported in literature (for examples, Table 5), the amount of the used catalysts for the same reaction conditions is around 30-100 mg. In our method, the mg of solid catalyst was just 6 mg!

 In many of the literature reports the optimized conditions are reached by studying one by one parameter and no correlation between the parameters can be screened. In our method, simultaneous effects of binary parameters were studied as can be seen in Figure 7. In this Figure, the dosage of the catalyst and NaBH4 have exact correlation in para-nitrophenol reduction, for example 0.002 M of NaBH4 for 0.002 g of catalyst but after 0.005 g of the catalyst the dosage of NaBH4 was not important factor. These kinds of binary parameters correlation curve also have novelty in our work.

The mentioned novelty parts in the main text and Figures (both manuscript and Electronic Supporting Information) were highlighted, previously.

Thank you again for your comment and helpful point of view on the improvements to our manuscript.

Reviewer 2 Report

Authors have addressed all comments and improved the manuscript.

Accepted in present form for publication.

Author Response

Thank you for your useful comments resulting in improving our work. We are thankful for accepting our manuscript.